# Prospective Evaluation of CD45RA+/CCR7- Effector Memory T (T_EMRA_) Cell Subsets in Patients with Primary and Secondary Brain Tumors during Radiotherapy of the Brain within the Scope of the Prospective Glio-CMV-01 Clinical Trial

**DOI:** 10.3390/cells12040516

**Published:** 2023-02-04

**Authors:** Ilka Scheer, Ina Becker, Charlotte Schmitter, Sabine Semrau, Rainer Fietkau, Udo S. Gaipl, Benjamin Frey, Anna-Jasmina Donaubauer

**Affiliations:** 1Translational Radiobiology, Department of Radiation Oncology, Universitätsklinikum Erlangen, Friedrich-Alexander-Universität Erlangen-Nürnberg (FAU), 91054 Erlangen, Germany; 2Department of Radiation Oncology, Universitätsklinikum Erlangen, Friedrich-Alexander-Universität Erlangen-Nürnberg (FAU), 91054 Erlangen, Germany; 3Comprehensive Cancer Center Erlangen-EMN, 91054 Erlangen, Germany

**Keywords:** T_EMRA_ cells, brain metastases, glioblastoma, cytomegalovirus (CMV), radiotherapy, infection

## Abstract

Radiotherapy (RT) of the brain is a common treatment for patients with high-grade gliomas and brain metastases. It has previously been shown that reactivation of cytomegalovirus (CMV) frequently occurs during RT of the brain. This causes neurological decline, demands antiviral treatment, and is associated with a worse prognosis. CMV-specific T cells are characterized by a differentiated effector memory phenotype and CD45RA+ CCR7- effector memory T (T_EMRA_) cells were shown to be enriched in CMV seropositive individuals. In this study, we investigated the distribution of T_EMRA_ cells and their subsets in the peripheral blood of healthy donors and, for the first time, prospectively within the scope of the prospective Glio-CMV-01 clinical trial of patients with high-grade glioma and brain metastases during radiation therapy as a potential predictive marker. First, we developed a multicolor flow cytometry-based assay to monitor the frequency and distribution of T_EMRA_ cells in a longitudinal manner. The CMV serostatus and age were considered as influencing factors. We revealed that patients who had a reactivation of CMV have significantly higher amounts of CD8+ T_EMRA_ cells. Further, the distribution of the subsets of T_EMRA_ cells based on the expression of CD27, CD28, and CD57 is highly dependent on the CMV serostatus. We conclude that the percentage of CD8+ T_EMRA_ cells out of all CD8+ T cells has the potential to serve as a biomarker for predicting the risk of CMV reactivation during RT of the brain. Furthermore, this study highlights the importance of taking the CMV serostatus into account when analyzing T_EMRA_ cells and their subsets.

## 1. Introduction

The cytomegalovirus (CMV) is a double-stranded DNA virus that belongs to the family of Herpesviridae. CMV is a widespread virus with a seroprevalence in adults of approximately 40–60% in industrialized nations. Even though the primary infection is usually asymptomatic in adults, symptomatic infections frequently occur in the setting of immunodeficiency, for example, in the setting of iatrogenic immunosuppression following organ transplantation. Various organs may be affected, leading to life-threatening infections in immunocompromised individuals. After primary infection, the virus persists in myeloid and endothelial cells, maintaining only a reduced expression of viral genes. However, reactivation occurs occasionally in immunocompromised individuals, such as elderly people, transplant patients, or patients undergoing (radio-) chemotherapy (RCT). Often, an antiviral treatment for those patients is indispensable [1,2,3].

It has previously been shown that reactivation of a latent CMV infection during radiation therapy (RT) of the brain can lead to encephalopathy with a sudden neurological decline in patients with high-grade gliomas and brain metastases [3]. Even though those patients did not have signs of tumor progression in MRI scans, they showed a neurologic decline, which is most likely due to a reactivation of CMV. This finding occurs in about 20% of those patients during or within 4 weeks after the start of RCT. Antiviral treatment reverses the neurologic symptoms and improves the quality of life in those patients. Still, the median overall survival of patients that developed CMV reactivation was significantly shorter compared to those without CMV reactivation. Due to the high incidence of CMV-associated neurological decline and its impact on prognosis, it is important to monitor glioblastoma patients for CMV reactivation. The prospective, observatory Glio-CMV-01 study (ClinicalTrials ID: NCT02600065) screens for reactivation of CMV and analyzes the immune status of patients with high-grade gliomas and brain metastases who are supposed to get radiation therapy of the brain. Within the study, a longitudinal flow cytometry-based immune monitoring (immunophenotyping) is performed at different time points during therapy to find possible prognostic biomarkers for complications associated with CMV. Low basophil counts before the start of RT have already been shown to predict a high risk for CMV-associated encephalopathy [2,3]. Nonetheless, the immunological mechanisms leading to CMV reactivation during RT have not been fully understood yet and RT of the brain is getting more and more sophisticated [4]. About 30 distinct immune cell subtypes and their respective activation state can be discriminated by immunophenotyping over the course of the therapy in the Glio-CMV-01 trial. Thus, a deeper insight into the immunological mechanisms of CMV reactivation in this patient collective is generated [5,6,7].

The immunophenotyping that had been done so far did not yet further differentiate T cell subsets. However, it has been shown that CMV seropositivity influences the distribution of T cell subsets towards a higher frequency of late differentiated T cells [8]. CD4+ and CD8+ T cells can be divided into four subsets by the surface markers CD45RA and CCR7. Those subsets are CD45RA+ CCR7+ naive T cells (T_N_), CD45RA- CCR7+ central memory T cells (T_CM_), CD45RA- CCR7- effector memory T cells (T_EM_), and CD45RA+ CCR7- effector memory T cells re-expressing CD45RA (T_EMRA_) [9,10]. T_EMRA_ cells are generally considered to be terminally differentiated cells due to their high cytotoxicity [11] and poor proliferation capacity [12]. The amount of both CD8+ and CD4+ T_EMRA_ cells is highly variable among individuals as it is influenced by factors such as age, genetic background, and immunological events during lifetime [13,14]. Different surface markers have been suggested to define further subsets of T_EMRA_ cells [15]. CD27- CD28- T_EMRA_ cells have been shown to have a later differentiated phenotype with high cytotoxicity, whereas the phenotype of CD27+ CD28-/+ T_EMRA_ seems to be intermediate between naïve and effector cells [16]. T_EMRA_ cells have been shown to be mostly CD27- CD28- [17]. Some authors use the senescence marker CD57 to define an even further differentiated, CD57+ T_EMRA_ subset [14,15]. Other authors combine a positive expression of CD57 with a negative expression of CD28 as a marker of senescence in T_EMRA_ cells [18,19]. 

It was previously shown that the CD8+ T_EMRA_ subset significantly increases with age. Still, this was not shown for CD4+ T_EMRA_ cells [17]. However, a more recent study did not find a correlation between age and the amount of T_EMRA_ cells in individuals older than 65 years [14]. Furthermore, CMV seropositive individuals have been found to have higher amounts of T_EMRA_ cells, both in the CD4+ and in the CD8+ compartment [20]. Nevertheless, there have also been studies that did not detect a significant correlation between the CMV serostatus and the amount of T_EMRA_ cells [21]. It was suggested that CMV might accelerate immunosenescence, which is defined as a decline in immune function associated with age. Immunosenescence results in a decreased resistance to infections [22]. CD8+ T_EMRA_ cells have a senescent phenotype, and their numbers seem to be increased in older individuals, CMV seropositive individuals, and those with chronic inflammatory diseases. Therefore CD8+ T_EMRA_ cells have been suggested as a biomarker for immunosenescence. Due to the connection between T_EMRA_ cells and immunosenescence [14], we hypothesized that T_EMRA_ cells and their subsets could either be relevant for the reactivation of CMV during radiation therapy of the brain or serve as a predictive marker. Therefore, this prospective study aimed to develop an assay for longitudinal immune monitoring and evaluated the distribution of T_EMRA_ cells and their subsets in patients with high-grade glioma and brain metastases during radiation therapy and healthy donors as a comparison. This work was performed within the scope of the Glio-CMV-01 clinical trial.

## 2. Materials and Methods

### 2.1. Study Design

The GLIO-CMV-01 study is a prospective observational study, conducted at the Department of Radiation Oncology of the Universitätsklinikum Erlangen in Germany. It is a registered clinical trial (ClinicalTrials ID: NCT02600065). The study was conducted in accordance with the Declaration of Helsinki and approved by the Institutional Review Board of the Friedrich-Alexander Universität Erlangen-Nürnberg (protocol code: 265_14B; date of approval: 18 September 2014). Informed consent was obtained from all subjects involved in the study.

Patients enrolled in the trial either suffered from high-grade gliomas (WHO grades III-IV) or brain metastases. The patients received local RT of the brain or whole-brain RT. The duration of local RT was 42 to 45 days and the duration of whole-brain RT was 14 to 28 days. Blood samples were taken before, in the middle of, and at the end of RT. Peripheral blood was tested for anti-CMV IgM, anti-CMV IgG, and CMV DNA. The testing was performed by the Institute of Virology of the Universitätsklinikum Erlangen immediately after the blood withdrawal. Viremia was defined as ≥250 copies/mL by real-time PCR. CMV-associated encephalopathy was considered proven if neurological decline occurred together with viremia. MRI scans were performed to exclude other explanations for the neurological decline. In the case of CMV-associated encephalopathy, patients were treated with ganciclovir or valganciclovir [2,3]. 

Simultaneously with the CMV analyses, whole blood immunophenotyping was performed as described in previously published protocols. Thereby, a detailed peripheral immune status of the patients was obtained throughout the study, covering nine main immune cell types and numerous subtypes and their respective activation status [5,6,7]. In the here presented analysis, we additionally determined T_EMRA_ cells and their subsets by multicolor flow cytometry in patients enrolled in the GLIO-CMV-01 study as well as in a group of healthy donors without tumor disease or radiation therapy. The inclusion of healthy donors was conducted in accordance with the Declaration of Helsinki and approved by the Institutional Review Board of the Friedrich-Alexander Universität Erlangen-Nürnberg (protocol code: 21-15-B; date of approval: 9 November 2022). Informed consent was obtained from all subjects involved in the study. All flow cytometric analyses were performed from whole blood without previous isolation of mononuclear cells. 

### 2.2. Patient Cohort

For this study, 21 healthy donors have been enrolled. From the Glio-CMV 01 study, a total number of 37 patients were enrolled. We performed the analysis of T_EMRA_ cells in 24 patients before the start of RT, in 23 patients halfway through, and in 19 patients at the end of RT. Since reactivation of CMV occurs in the setting of immunodeficiency [1] and none of the included healthy donors is suffering from any kind of immunosuppression, the analysis of T_EMRA_ cells and the determination of the CMV serostatus in the healthy donors was performed only once. Due to the above-described influence on the amount of T_EMRA_ cells, the patient cohort as well as the healthy donors were each divided into two groups depending on the CMV serostatus. In the CMV seropositive group, CMV reactivation, which was diagnosed by the above-described criteria, occurred in 3 patients. The detailed characteristics of the healthy donors are listed in Table 1 and the patient cohort is described in Table 2.

### 2.3. Flow Cytometric Analysis

The following antibodies were applied for the determination of T_EMRA_ cells: CD3-Krome Orange (Beckman Coulter, Brea, CA, USA, Cat# B00068, RRID:AB_2892547), CD8-FITC (BD Biosciences, Franklin Lakes, NJ, USA, Cat# 555634, RRID:AB_395996), CD4-PerCP Cy5.5 (BD Biosciences, Franklin Lakes, NJ, USA, Cat# 560650, RRID:AB_1727476), CD45RA-PE Cy7 (BD Biosciences, Franklin Lakes, NJ, USA, Cat# 560675, RRID:AB_1727498), CCR7-PE (BD Biosciences, Franklin Lakes, NJ, USA, Cat# 560765, RRID:AB_2033949), CD27-BV 421 (BD Biosciences, Franklin Lakes, NJ, USA, Cat# 742731, RRID:AB_2741005), CD28-APC (BD Biosciences, Franklin Lakes, NJ, USA, Cat# 560683, RRID:AB_1727458), and CD57-Brilliant Violet 605 (BioLegend, San Diego, CA, USA, Cat# 393303, RRID:AB_2728425). Blood samples were collected in EDTA tubes and analyzed within 24 h. Each sample was analyzed following the same standardized procedure. For each measurement, 100 μL of whole blood were incubated with 5 μL of CD3-Krome Orange, 4 μL of CD8-FITC, 1 μL of CD4-PerCP Cy5.5, 2.5 μL of CD45RA-PE Cy7, 10 μL of CCR7-PE, 0.5 μL of CD27-BV 421, 2.5 μL of CD28-APC, 10 μL of CD57-Brilliant Violet 605, and 14.5 μL of PBS in a 5 mL polypropylene FACS tube at room temperature for 20 min in the dark. In the following, lysis of the erythrocytes and fixation of the leukocytes were performed by a TQ Prep Workstation (Beckman Coulter, Brea, CA, USA). The samples were washed twice with phosphate-buffered saline (PBS) (Sigma-Aldrich, München, Germany). A total of 3 mL of PBS was added to the FACS tube and the cells were centrifuged at 300× *g* for 5 min. The fluid supernatant was carefully removed so that only fixated leukocytes remained. After the second washing step, the cells were dissolved in 200 μL of PBS (Sigma-Aldrich, München, Germany). For further information, please refer to previous publications on the procedure of flow cytometric immunophenotyping which were published by our research group. [5,6,7].

Data acquisition was immediately performed on a Gallios flow cytometer (Beckman Coulter, Brea, CA, USA) in the standard filter setting. Analysis was performed using Kaluza Analysis Software (Version 2.1, Beckman Coulter, Brea, CA, USA). The gating strategy is depicted in Figure 1.

### 2.4. Statistical Analysis

Microsoft Excel 2016 (Microsoft Corporation, Redmont, WA, USA) was used for data management and GraphPad Prism 9 (GraphPad Software Inc., La Jolla, CA, USA) was used for the statistical analysis. The non-parametric two-tailed Mann–Whitney U-test was applied to assess the differences between the percentages of the T cell subsets. A Kruskal–Wallis test was used to compare the amount of T_EMRA_ cells between the different time points of radiation therapy. A Spearman correlation and simple linear regression were performed to correlate the percentage of T_EMRA_ cells to the age of the included healthy donors and patients. A *p*-value < 0.05 was considered statistically significant.

## 3. Results

### 3.1. CMV Seropositive Healthy Individuals Have Significantly Higher Amounts of CD8+ T_EMRA_ Cells

First, we analyzed the distribution of both CD8+ and CD4+ T cells into T_N_, T_CM_, T_EM,_ and T_EMRA_ cells in healthy donors, as described in the gating strategy in Figure 1. In accordance with previous studies [20], we found a significantly higher amount of CD8+ T_EMRA_ cells and a significantly lower amount of CD8+ T_N_ cells in CMV seropositive individuals compared to CMV seronegative individuals. There are no significant differences in the distributions of CD8+ T_CM_ and CD8+ T_EM_. (Figure 2A). Considering CD4+ T cells, we did not find any significant differences between CMV seropositive and seronegative individuals. However, there is also a tendency towards a lower amount of T_N_ cells and a higher amount of T_EMRA_ cells in CMV seropositive individuals (Figure 2B).

### 3.2. The Amount of T_EMRA_ Cells during Radiation Therapy

Next, we compared the percentages of CD8+ and CD4+ T_EMRA_ cells before treatment, halfway through, and after RT in patients to see whether RT induces changes in the percentage of T_EMRA_ cells in the T cell compartment of the peripheral blood. However, there are no significant differences between the different time points of evaluation (Figure 3). However, as observed in the healthy donors, higher CD8+ T_EMRA_ cells are present in the CMV seropositive tumor patients, which was most pronounced halfway through RT (Figure 3A). In contrast to the healthy donors, CD4+ T_EMRA_ cells were significantly increased, particularly halfway through and after RT (Figure 3B).

### 3.3. Patients with CMV Reactivation Have a Significantly Higher Percentage of CD8+ T_EMRA_ Cells

Further, we were interested, in whether the frequency of T_EMRA_ cells is different in brain tumor patients, especially during reactivation of CMV, in comparison to healthy donors. We compared the percentage of CD8+ and CD4+ T_EMRA_ cells out of the total CD8+ and CD4+ T cells between CMV seropositive healthy donors, CMV seropositive patients before the start of RT that did not develop reactivation of CMV, and patients with a diagnosed CMV reactivation according to the above-described criteria. The percentage of CD8+ T_EMRA_ cells in patients who had a reactivation of CMV is significantly higher than in patients who did not have a reactivation of CMV. Also, the percentage of CD8+ T_EMRA_ cells in patients with a CMV reactivation is significantly higher than in healthy donors. The median percentage of CD8+ T_EMRA_ cells in patients without a reactivation is higher than in healthy individuals, but the difference does not reach statistical significance (Figure 4A). The amount of CD4+ T_EMRA_ cells is not significantly different between patients who had a CMV reactivation and healthy donors or patients without a reactivation. However, it was noticeable that one patient with a CMV reactivation had an unusually high amount of CD4+ T_EMRA_ cells (55.3% of CD4+ T cells; Figure 4B).

### 3.4. Subsets of T_EMRA_ Cells Are Strongly Influenced by the CMV Serostatus

To generate further insight into the distribution of T_EMRA_ cells and their respective subsets in peripheral blood, we divided the CD4+ and CD8+ T_EMRA_ cells into subpopulations based on their expression of CD27 and CD28 in healthy donors and patients. There are strong differences between CMV seropositive and CMV seronegative individuals in the distribution of those subsets. In the CD8+ compartment, the amount of CD28- CD27- T_EMRA_ cells, which was shown to be a later differentiated phenotype of T_EMRA_ cells [16], is significantly lower in CMV seronegative individuals in both the healthy group (Figure 5A) as well as in the patient group (Figure 5B). In CMV seropositive healthy donors and patients, the CD28- CD27- subset is the largest subset within CD8+ T_EMRA_ cells. The amounts of CD28- CD27+ and CD28+ CD27+ T_EMRA_ cells are significantly higher in CMV seronegative healthy donors and patients (Figure 5A,B).

In the CD4+ compartment, as detected for CD8+ cells, the amount of CD28- CD27- T_EMRA_ cells is significantly lower in CMV seronegative healthy donors (Figure 5C) and patients (Figure 5D) compared to CMV seropositive individuals. Further, the amount of CD27+ CD28+ T_EMRA_ cells is significantly higher in CMV seronegative healthy donors and patients compared to CMV seropositive individuals. In contrast to the CD8 compartment, the amount of CD28- CD27+ T_EMRA_ cells is significantly lower in CMV seronegative healthy donors compared to CMV seropositive healthy donors, while in tumor patients only a tendency of decreased amounts is observed (Figure 5C,D).

Furthermore, we divided T_EMRA_ cells into four more subsets based on the expression of CD28 and CD57. Those subsets are also highly different between CMV seropositive and CMV seronegative individuals. Referring to the CD8+ T_EMRA_ cells, the amount of CD28+ CD57- T_EMRA_ cells is significantly higher in CMV seronegative healthy donors (Figure 6A) and patients (Figure 6B). The amount of CD28- CD57- T_EMRA_ cells is significantly lower in seronegative healthy donors and patients (Figure 6A,B).

Considering the CD4+ T_EMRA_ cells, in both the patient and the healthy group, the CD28- CD57+ subset was hardly measurable in seronegative individuals, whereas in seropositive individuals, we found varying numbers of up to 88.4% of CD4+ T_EMRA_ cells. Moreover, in CMV seronegative patients and healthy donors, the amount of CD28- CD57- T_EMRA_ cells is significantly lower, and the amount of CD28+ CD57- T_EMRA_ cells is significantly higher compared to seropositive individuals (Figure 6C,D). 

### 3.5. Influence of Age on the Distribution of T_EMRA_ Cells

Since age has also been described as a factor influencing the amount of T_EMRA_ cells [17], we correlated the percentages of CD8+ and CD4+ T_EMRA_ cells out of all CD8+ and CD4+ T cells to the age of healthy donors and patients before the start of RT (Figure 7). 

In the cohorts included in this study, we did not find a significant correlation between age and the percentage of CD8+ or CD4+ T_EMRA_ cells (Figure 7).

## 4. Discussion

In this study, we first developed a flow cytometry assay for the detection of T_EMRA_ cell subsets in the peripheral blood that can be complementarily used as an addition to the classical flow cytometric immunophenotyping previously published [5,6,7]. To our knowledge, this was the first study to analyze T_EMRA_ cells in a group of patients with high-grade brain tumors or brain metastases at different time points during RT of the brain. Irradiation techniques and concepts for the treatment of malignant lesions in the brain have been improved strongly in recent years and are becoming more and more targeted [23].

First, we confirmed previous findings that healthy CMV seropositive individuals have a significantly higher percentage of CD8+ T_EMRA_ cells compared to CMV seronegative individuals [20]. It is believed that, because of the capacity of CMV to reactivate, the human immune system is constantly challenged to control the virus, which leads to a different distribution of T cell subsets in CMV seropositive individuals. Infection with CMV causes a reduction in the number of naïve T cells and a higher number of T cells with a later differentiated phenotype [8,20]. It has been shown that a high proportion of CD4+ and CD8+ T cells in the peripheral blood of CMV seropositive individuals responds specifically to CMV [24]. However, the exact mechanism behind those changes in the immune system remains unclear [25].

In this study, the percentage of CD4+ T_EMRA_ cells is not significantly higher in healthy CMV seropositive individuals compared to CMV seronegative individuals. However, a tendency towards a higher percentage of CD4+ T_EMRA_ cells in CMV seropositive healthy donors can be seen. Although some previous studies did not find higher CD4+ T_EMRA_ cells in CMV seropositive individuals, many other studies did [21]. We assume that we only see a tendency towards higher CD4+ T_EMRA_ cells in CMV seropositive healthy individuals but no significant difference due to the small sample size and the rather low percentage of CD4+ T_EMRA_ cells in general. Furthermore, we found a high interindividual variation in the percentages of CD8+ and CD4+ T_EMRA_ cells, which is in accordance with previous studies on T_EMRA_ cells. Burel, J.G., et al. described CD4+ T_EMRA_ cells ranging from less than 0.3% to about 18% of CD4+ T cells and CD8+ T_EMRA_ cells ranging from about 4% to 67% of CD8+ T cells in a healthy population [13]. The accordance of our results with previous findings in the literature proves that the developed flow cytometry assay for T_EMRA_ detection is robust. 

Although we did not find a significant difference between CMV seropositive and CMV seronegative CD4+ T_EMRA_ cells in the patient group before the start of RT, there is a significant difference between CMV seronegative and CMV seropositive individuals halfway through and at the end of RT. Considering the CD8+ T_EMRA_ cells, there is a significant difference halfway through RT. The percentage of CD8+ T_EMRA_ cells in CMV seropositive patients tends to be slightly higher halfway through RT compared to the percentage at the beginning and at the end of RT. However, we showed that there are no significant differences in the percentages of T_EMRA_ cells between the first time point compared to the other two time points during RT. Previously, Burel, J.G., et al. also described a low intraindividual variability in the amount of CD8+ T_EMRA_ cells in healthy individuals over the course of a few months, suggesting that they are relatively stable [13]. We could confirm this finding not only for CD8+ T_EMRA_ cells but also for CD4+ T_EMRA_ cells. Falcke, S.E., et al. showed that direct radiation of T cells with a single dose of ≥2.0 Gy strongly reduces T cell viability. The T cells mainly died by necrosis and their sensitivity to radiation was considered moderate [26]. In our study, radiation therapy of the brain apparently did not affect the CD8+ and CD4+ T_EMRA_ cells in the peripheral blood. In the future, it might be interesting to determine the sensitivity of isolated T_EMRA_ cells to direct radiation because Falcke, S.E., et al. did not differentiate between T cell subsets. Since it was previously described that one aspect of the highly differentiated phenotype of T_EMRA_ cells is a high sensitivity to apoptosis [12,15], their radiosensitivity might be higher than the radiosensitivity of T cells in general.

We showed for the first time that the percentage of CD8+ T_EMRA_ cells is significantly higher in patients that had a reactivation of CMV compared to CMV seropositive patients without a reactivation and to CMV seropositive healthy individuals. In fact, the patients with a CMV reactivation have a higher mean age than the CMV seropositive patients in general (78.3 versus 65.1 years), and there have been some studies that found a positive correlation between age and the amount of CD8+ T_EMRA_ cells [17]. However, in the population of patients that we analyzed, there is no significant correlation between age and the amount of T_EMRA_ cells in CMV seropositive patients and healthy individuals. Salumets, A., et al. also did not find a correlation between age and the amount of CD8+ T_EMRA_ cells in individuals older than 65 years of age [14]. Therefore, it seems unlikely to us that the higher percentage of CD8+ T_EMRA_ cells is a mere effect of the higher mean age in this group. Another explanation could be that patients with a higher percentage of CD8+ T_EMRA_ cells have specific changes in their immune system which make them more likely to develop reactivation of CMV during RT of the brain. Macaulay, R., et al. described that the upregulation of proinflammatory molecules, which are amongst others produced by T_EMRA_ cells, drives CMV reactivation [27], and Salumets, A., et al. suggested CD8+ T_EMRA_ cells as a biomarker for immunosenescence [14]. Therefore, it seems likely that a high level of CD8+ T_EMRA_ cells might indicate a higher risk for CMV reactivation in patients receiving RT of the brain. However, longitudinal monitoring of T_EMRA_ cells in a larger group of patients is necessary to confirm this hypothesis. It must be considered that even though we saw a significant difference, the number of three patients is quite small, and therefore should only be seen as a first hint which might indicate a direction to explore in future studies on a larger group of patients. Goerig et al. described that reactivation of CMV with neurologic decline occurs in about 20% of patients with brain tumors during or within 4 weeks after the start of RCT [2,3]. In the patient population that we analyzed, CMV reactivation was diagnosed in three out of 16 CMV seropositive patients, which is about 18.8%.

Previously, a higher amount of CD8+ TEMRA cells has already been shown to correlate not only with CMV seropositivity [20] but also with other comorbidities or adverse health events, such as end-stage renal disease [28], a higher risk for graft failure after kidney transplant [29], or a higher risk for cardiovascular mortality [30]. We did not find a significant difference between the percentage of CD4+ T_EMRA_ in patients who had a CMV reactivation compared to patients who did not have a reactivation. The one patient with an unusually high percentage of CD4+ T_EMRA_ cells might be an outlier. Since CD8+ T_EMRA_ cells are found at higher percentages than CD4+ T_EMRA_ cells [13], differences are easier to measure in the CD8+ compartment. Another explanation might be that, in general, CD8+ T cells are mostly cytolytic and they are able to directly destroy virus-infected cells during antigen-specific contact. This is why they are especially important for the defense against viral infections [31], and therefore effects might rather be seen in the CD8+ T cell compartment.

Referring to the subsets of T_EMRA_ cells based on the expression of the surface markers CD27, CD28, and CD57, we revealed a strong difference in the distribution of T_EMRA_ cell subsets between CMV seronegative and CMV seropositive individuals in the healthy group as well as in the patient group. The percentage of the CD27- CD28- subset, which is considered to have a later differentiated phenotype and a high cytotoxicity [16], is significantly higher in CD8+ and CD4+ T_EMRA_ cells in CMV seropositive healthy individuals as well as in patients. The percentage of the differentiated CD57+ CD28- subset [18,19] is significantly higher in CD4+ T_EMRA_ cells in CMV seropositive healthy individuals and in patients. Although there is no significant difference concerning the CD8+ T_EMRA_ cells, there is a tendency towards a higher amount of CD57+ CD28- T_EMRA_ cells in CMV seropositive healthy donors and patients. CD27+ T_EMRA_ cells are considered to have an intermediate phenotype between naïve and effector cells [16]. The percentage of CD27+ CD8+ T_EMRA_ cells is significantly lower in CMV seropositive healthy donors and patients. Considering CD4+ T_EMRA_ cells, the percentage of CD27+ CD28+ T_EMRA_ cells is lower in CMV seropositive healthy donors and patients, but, interestingly, the percentage of CD27+ CD28- T_EMRA_ cells is higher in CMV seropositive healthy donors in this study. It has been suggested before that CD4+ T_EMRA_ cells might be heterogenous in terms of differentiation states, with some of them being similar to T_EM_ cells and others expressing surface markers associated with terminal differentiation [32]. Altogether, our findings are in accordance with previous studies, which describe that CMV seropositivity is associated with a shift of immune cell subsets towards highly differentiated subsets [14]. The differences between CMV seropositive and CMV seronegative individuals in the T_EMRA_ cell subsets are more pronounced than the differences between the percentages of CD8+ and CD4+ T_EMRA_ cells in general. In our view, these results demonstrate the importance of determining the CMV serostatus before analyzing T_EMRA_ cells and their respective subsets.

Our study provides novel insights into the distribution and possible role of CD8+ and CD4+ T_EMRA_ cells and their subsets in healthy donors and patients with high-grade gliomas and brain metastases receiving RT. In the future, monitoring of T_EMRA_ cells in even larger patient cohorts has to be performed to elucidate whether distinct CD8+ T_EMRA_ cells could serve as a biomarker in patients at risk of developing CMV reactivation with possibly fatal consequences during RT of the brain. A therapeutic consequence that could be considered is the administration of antiviral prophylaxis for patients with a high risk for a CMV reactivation similar to the antiviral prophylaxis which is routinely performed for patients after solid organ transplantation [33]. However, possible side effects of the antiviral treatment must be weighed against the potential benefit of preventing a reactivation. It might also be interesting to characterize the functional phenotype of T_EMRA_ cells in patients with high-grade gliomas and brain metastases to get a better understanding of their role in patients that develop reactivation of CMV.

## Figures and Tables

**Figure 1 cells-12-00516-f001:**
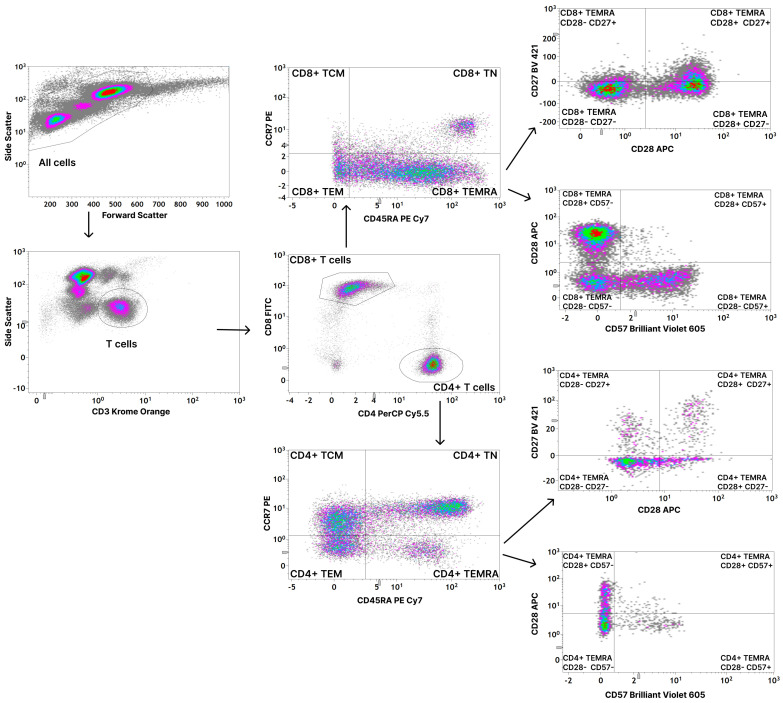
Gating strategy for the assessment of CD8+ and CD4+ CD45RA+ CCR7- effector memory T cells (T_EMRA_) and their subsets. T cells were gated as CD3+ leukocytes and divided into CD8+ and CD4+ T cells. Both CD8+ and CD4+ T cells were divided into naive T cells (T_N_), central memory T cells (T_CM_), effector memory T cells (T_EM_), and effector memory T cells re-expressing CD45RA (T_EMRA_) based on the expression of CD45RA and CCR7. T_EMRA_ cells were defined as CD45RA+ CCR7-. Both CD8+ and CD4+ T_EMRA_ cells were divided into further subsets based on the expression of CD27 and CD28 and on the expression of CD28 and CD57. Gates for CCR7, CD45RA, CD27, and CD57 were set using fluorescence minus one (FMO) controls.

**Figure 2 cells-12-00516-f002:**
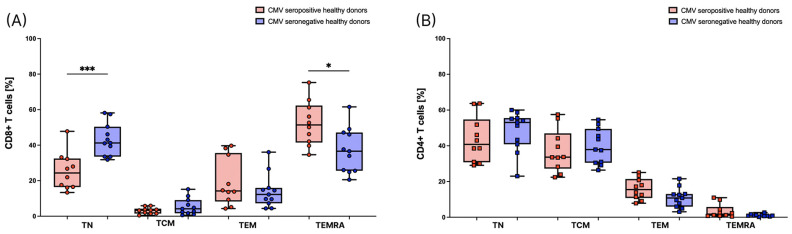
Distribution of T cell subsets in healthy donors. T_N_, T_CM_, T_EM,_ and T_EMRA_ cells are depicted as percentages of CD8+ T cells in (**A**), and of CD4+ T cells in (**B**). Individuals with cytomegalovirus (CMV) seropositivity are depicted in red (*n* = 10) and individuals that are seronegative for CMV are depicted in blue (*n* = 11). For statistical analysis, a non-parametric two-tailed Mann–Whitney U-test was used (*: *p* < 0.05; ***: *p* < 0.001).

**Figure 3 cells-12-00516-f003:**
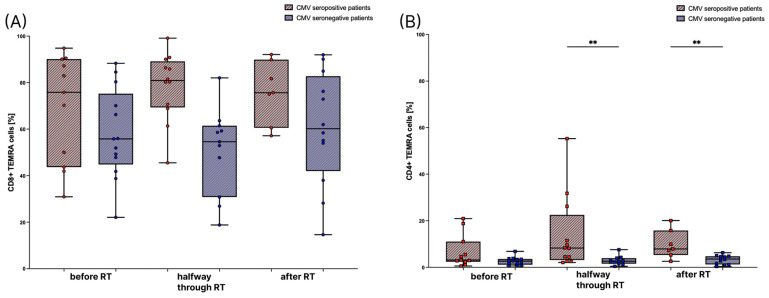
Amounts of T_EMRA_ cells before, during, and after radiotherapy (RT). T_EMRA_ cells are depicted as percentages of CD8+ T cells in (**A**), and of CD4+ T cells in (**B**). T_EMRA_ cells were measured in CMV seropositive patients, depicted in red, before RT (*n* = 11), halfway through RT (*n* = 12), and after RT (*n* = 7), as well as in CMV seronegative patients, depicted in blue, before RT (*n* = 13), halfway through RT (*n* = 11), and after RT (*n* = 12). For statistical analysis, the Kruskal–Wallis test was used to compare the amount of T_EMRA_ cells halfway through and after RT to the amount before RT. For the analysis between the CMV seronegative and seropositive group within each time point, a non-parametric two-tailed Mann–Whitney U-test was applied (**: *p* < 0.01).

**Figure 4 cells-12-00516-f004:**
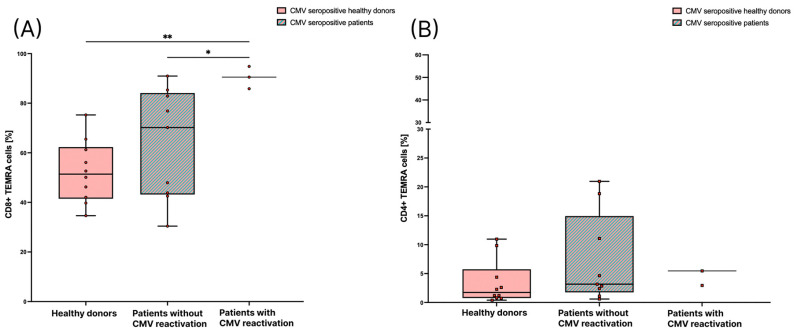
Comparison of T_EMRA_ cells in healthy donors and patients with or without CMV reactivation. T_EMRA_ cells are depicted as percentages of CD8+ T cells in (**A**), and of CD4+ T cells in (**B**). T_EMRA_ cells were measured in CMV seropositive healthy donors (*n* = 10), CMV seropositive patients before radiation therapy without reactivation (*n* = 9), and patients in the CMV seropositive group who had a reactivation of CMV (*n* = 3). For the statistical analysis, a non-parametric two-tailed Mann–Whitney U-test was applied (*: *p* < 0.05; **: *p* < 0.01).

**Figure 5 cells-12-00516-f005:**
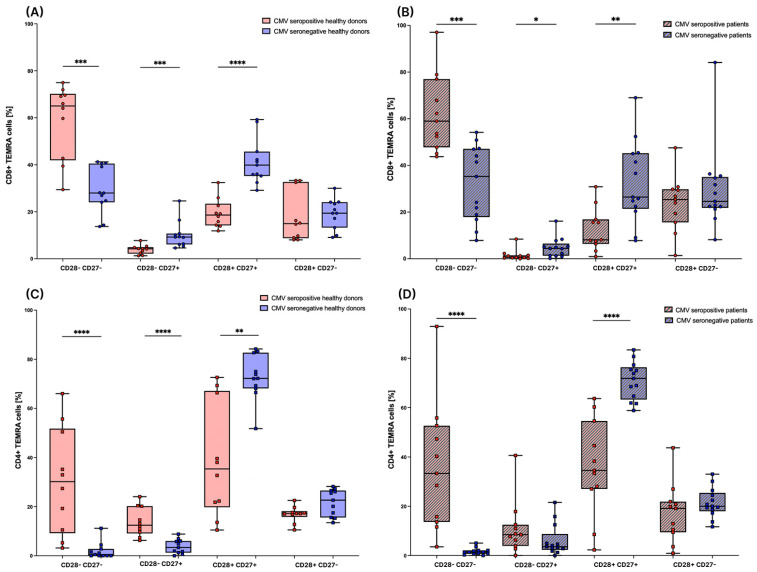
T_EMRA_ cell subsets based on the expression of CD27 and CD28. (**A**): CD8+ T_EMRA_ cells in healthy donors were divided into four subsets based on the expression of CD27 and CD28. T_EMRA_ cell subsets are depicted as percentages of CD8+ T_EMRA_ cells. Individuals with CMV seropositivity are depicted in red (*n* = 10) and individuals that are seronegative for CMV are depicted in blue (*n* = 11). (**B**): CD8+ T_EMRA_ cells in patients before RT were divided into four subsets based on the expression of CD27 and CD28. They are depicted as percentages of CD8+ T_EMRA_ cells. Individuals with CMV seropositivity are depicted in red (*n* = 11) and individuals that are seronegative for CMV are depicted in blue (*n* = 13). (**C**): CD4+ T_EMRA_ cells in healthy donors were divided into four subsets based on the expression of CD27 and CD28. They are depicted as percentages of CD4+ T_EMRA_ cells. Individuals with CMV seropositivity are depicted in red (*n* = 10) and individuals that are seronegative for CMV are depicted in blue (*n* = 11). (**D**): CD4+ T_EMRA_ cells in patients before RT were divided into four subsets based on the expression of CD27 and CD28. They are depicted as percentages of CD4+ T_EMRA_ cells. Individuals with CMV seropositivity are depicted in red (*n* = 11) and individuals that are seronegative for CMV are depicted in blue (*n* = 13). For the statistical analysis, a non-parametric two-tailed Mann–Whitney U-test was used (*: *p* < 0.05; **: *p* < 0.01; ***: *p* < 0.001; ****: *p* < 0.0001).

**Figure 6 cells-12-00516-f006:**
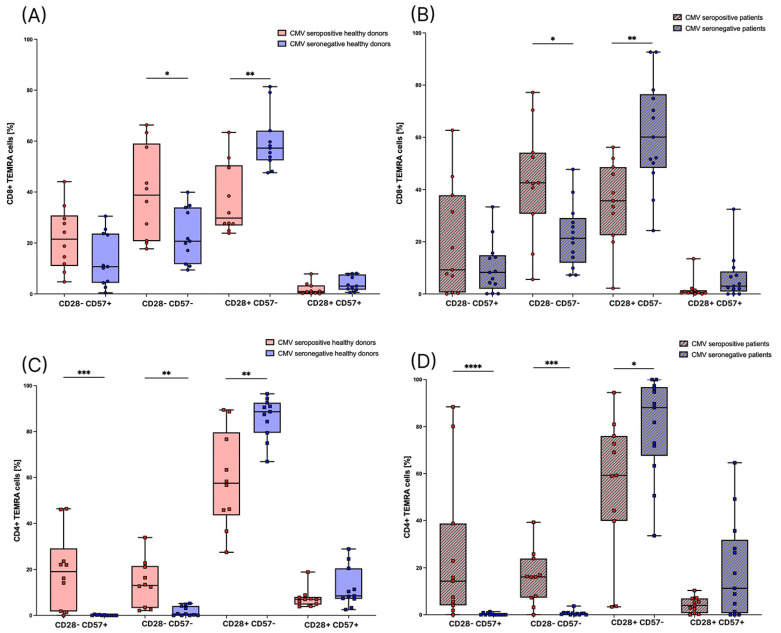
T_EMRA_ cell subsets based on the expression of CD57 and CD28. (A): CD8+ T_EMRA_ cells in healthy donors were divided into four subsets based on the expression of CD57 and CD28. They are depicted as percentages of CD8+ T_EMRA_ cells. Individuals with CMV seropositivity are depicted in red (*n* = 10) and individuals that are seronegative for CMV are depicted in blue (*n* = 11). (**B**): CD8+ T_EMRA_ cells in patients before RT were divided into four subsets based on the expression of CD57 and CD28. They are depicted as percentages of CD8+ T_EMRA_ cells. Individuals with CMV seropositivity are depicted in red (*n* = 11) and individuals that are seronegative for CMV are depicted in blue (*n* = 13). (**C**): CD4+ T_EMRA_ cells in healthy donors were divided into four subsets based on the expression of CD57 and CD28. They are depicted as percentages of CD4+ T_EMRA_ cells. Individuals with CMV seropositivity are depicted in red (*n* = 10) and individuals that are seronegative for CMV are depicted in blue (*n* = 11). (**D**): CD4+ T_EMRA_ cells in patients before RT were divided into four subsets based on the expression of CD57 and CD28. They are depicted as percentages of CD4+ T_EMRA_ cells. Individuals with CMV seropositivity are depicted in red (*n* = 11) and individuals that are seronegative for CMV are depicted in blue (*n* = 13). For the statistical analysis, a non-parametric two-tailed Mann–Whitney U-test was used (*: *p* < 0.05; **: *p* < 0.01; ***: *p* < 0.001; ****: *p* < 0.0001).

**Figure 7 cells-12-00516-f007:**
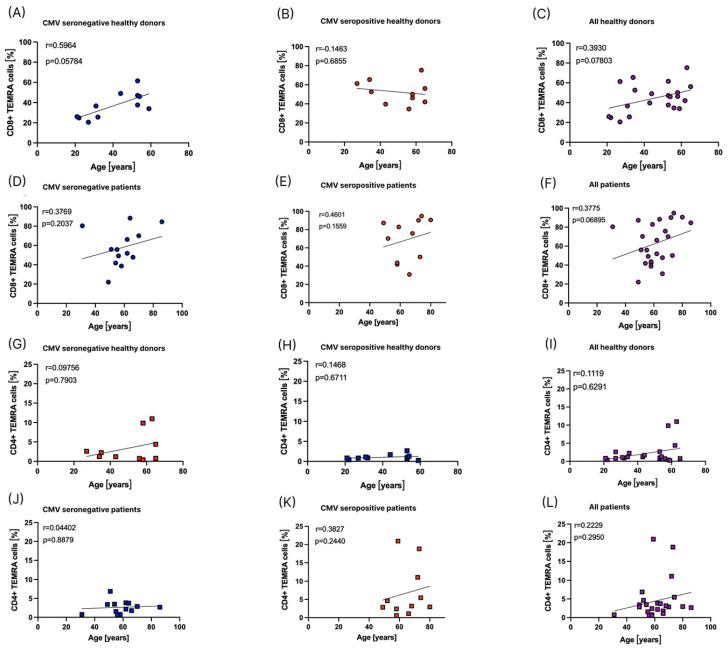
Influence of age on the percentage of T_EMRA_ cells. A Spearman correlation and a simple linear regression were performed to correlate the percentage of CD8+ T_EMRA_ cells out of all CD8+ T cells to age, depicted as circles, (**A**–**F**) and the percentage of CD4+ T_EMRA_ cells out of all CD4+ T cells to age, depicted as squares, (**G**–**L**). CMV seronegative individuals are depicted in blue, CMV seropositive individuals are depicted in red, and CMV seronegative and seropositive individuals combined are depicted in purple. (**A**) shows CMV seronegative healthy donors (*n* = 11). (**B**) shows CMV seropositive healthy donors (*n* = 10). (**C**) shows all healthy donors independent of CMV serostatus (*n* = 21). (**D**) shows CMV seronegative patients before the start of RT (*n* = 13). (**E**) shows CMV seropositive patients before the start of RT (*n* = 11). (**F**) shows all patients before the start of RT independent of CMV serostatus (*n* = 24). (**G**) shows CMV seronegative healthy donors (*n* = 11). (**H**) shows CMV seropositive healthy donors (*n* = 10). (**I**) shows all healthy donors independent of CMV serostatus (*n* = 21). (**J**) shows CMV seronegative patients before the start of RT (*n* = 13). (**K**) shows CMV seropositive patients before the start of RT (*n* = 11). (**L**) shows all patients before the start of RT independent of CMV serostatus (*n* = 24).

**Table 1 cells-12-00516-t001:** Healthy donor characteristics.

Factor	Category	CMV Seronegative (*n* = 11)	CMV Seropositive (*n* = 10)
Age at start	Mean Range	40.8	50.1
		21–59	27–65
Gender	Male Female	2 (18.2%)	1 (10%)
		9 (81.8%)	9 (90%)

**Table 2 cells-12-00516-t002:** Patient characteristics.

Factor	Category	CMV Seronegative (*n* = 21)	CMV Seropositive (*n* = 16)
Age at start	Mean Range	57.9	65.1
		27–86	38–81
Gender	Male Female	17 (81%)	8 (50%)
		4 (19%)	8 (50%)
Tumor disease	High-grade brain tumor (WHO III-IV) Brain metastases	18 (85.7%) 3 (14.3%)	13 (81.2%) 3 (18.8%)
Radiotherapy (RT)	Whole Brain Radiation Local Radiation	2 (9.5%) 19 (90.5%)	2 (12.5%) 14 (87.5%)
Single dose of RT	Mean Range	2.2 Gy 1.8–4 Gy	2.3 Gy 1.8–4 Gy
Cumulative dose of RT	Mean Range	52.7 Gy 12–60 Gy	51.7 Gy 30–60 Gy
Chemotherapy during RT	No Yes	4 (19%) 17 (81%)	3 (18.8%) 13 (81.2%)
Immunotherapy during RT	No Yes	21 (100%) 0 (0%)	14 (87.5%) 2 (12.5%)
Reactivation of CMV	No Yes	21 (100%) 0 (0%)	13 (81.2%) 3 (18.8%)

## Data Availability

The data presented in this study are available, on reasonable request, from the corresponding author.

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
