# Peer review of "Prospective Evaluation of CD45RA+/CCR7- Effector Memory T (TEMRA) Cell Subsets in Patients with Primary and Secondary Brain Tumors during Radiotherapy of the Brain within the Scope of the Prospective Glio-CMV-01 Clinical Trial"

_cells, 2023, doi:10.3390/cells12040516_

Round 1

Reviewer 1 Report

This study investigated the T cell subsets in patients with seropositive and negative for CMV.

I found there were many unclear points in this study.

Introduction (P2, L98-99). “It was suggested that CMV might accelerate immunosenescence.” Why are you discussing this? Please clarify.

Introduction (P3, L103). “Due to their above-mentioned characteristics, …” Please clarify what characteristics.

Introduction (P3, L105-109). These sentences are just methods. Please provide the purpose of this study. In addition, is this a clinical trial study? It should be in the title.

Discussion (P13, L404-406). There were only 3 CMV reactivations. The authors should discuss this limitation more. The authors compared seronegative and seropositive. Is CMV reactivation involved in this comparison?

Was CMV reactivation or T cell subsets presented in this study associated with poor prognosis?

Reviewer 2 Report

Review

Prospective evaluation of CD45RA+/CCR7- effector memory T 2 (TEMRA) cell subsets in patients with primary and secondary 3 brain tumors during radiotherapy of the brain.

 Ilka Scheer 1,2 , Ina Becker 1,2 , Charlotte Schmitter2,3 , Sabine Semrau2,3 , Rainer Fietkau2,3 , Udo S. Gaipl1,2,3* , Benjamin 5 Frey1,2,3 # and Anna-Jasmina Donaubauer1,2,3

I would like to congratulate the authors on exploring this prognostically and for the patient’s quality of life important, however, in the daily clinical radiotherapy routine more or less neglected issue.

I have read this manuscript with great interest and, even on repeated reading, found only very few issues where a further improvement of the quality could be suggested. They are, in any case, mostly of minor character.

Methods:

Line 112:  I would suggest to exchange the word ‘performed’ for ‘conducted’.

Line 121: Since the RT was conducted in the past, I suggest to use ‘… duration … was …’ instead of ‘… duration … is …’

Lines 146/147:

‘The analysis of TEMRA cells and the determination of the CMV serostatus in the 146 healthy donors was performed only once.’

Question: What is the likelihood that a reactivation occurs in CMV positive healthy donors in the time period required for the entire course of radiotherapy?

Table 2:

In the rows informing about chemo- and immunotherapy, it would be more informative if we learned how many percent of initially CMV positive patients experienced a reactivation. Since there are only three reactivation cases altogether, the distribution between patients who received additional Chemotherapy and those who did has no statistical power anyway.

Lines 166-168:

Should the volume  or concentration of each antibody in the final mix be mentioned?

Line 189:

Please, insert the statistical methods / tests used.

Discussion:

Line 341: Perhaps ‘The latter …’ could be replaced by ‘ Irradiation techniques and concepts for the treatment of malignant lesions in the brain …’?

Line 345: A comma is missing after ‘It is believed that’.

Curiosity question:

Would it have a therapeutic consequence if one could point out the CMV positive patients at risk early enough? Would the chemotherapy be reconsidered or would the patient receive some type of vaccination or boost of the immune system?

Round 2

Reviewer 1 Report

 I thank the authors for addressing all my comments. The manuscript has been greatly improved and I recommend its publication.